# Convolutional Complex Knowledge Graph Embeddings

Caglar Demir and Axel-Cyrille Ngonga Ngomo

Data Science Research Group, Paderborn University

**Abstract.** We investigate the problem of learning continuous vector representations of knowledge graphs for predicting missing links. Recent results suggest that using a Hermitian inner product on complex-valued embeddings or convolutions on real-valued embeddings can be effective means for predicting missing links. We bring these insights together and propose CONEX—a multiplicative composition of a 2D convolution with a Hermitian inner product on complex-valued embeddings. CONEX utilizes the Hadamard product to compose a 2D convolution followed by an affine transformation with a Hermitian inner product in $\mathbb{C}$. This combination endows CONEX with the capability of (1) controlling the impact of the convolution on the Hermitian inner product of embeddings, and (2) degenerating into ComplEx if such a degeneration is necessary to further minimize the incurred training loss. We evaluated our approach on five of the most commonly used benchmark datasets. Our experimental results suggest that CONEX outperforms state-of-the-art models on four of the five datasets w.r.t. Hits@1 and MRR even without extensive hyperparameter optimization. Our results also indicate that the generalization performance of state-of-the-art models can be further increased by applying ensemble learning. We provide an open-source implementation of our approach, including training and evaluation scripts as well as pretrained models.[1]

## 1 Introduction

Knowledge Graphs (KGs) represent structured collections of facts modelled in the form of typed relationships between entities [13]. These collections of facts have been used in a wide range of applications, including web search [10], cancer research [29], and even entertainment [21]. However, most KGs on the Web are far from being complete [24]. For instance, the birth places of 71% of the people in Freebase and 66% of the people in DBpedia are not found in the respective KGs. In addition, more than 58% of the scientists in DBpedia are not linked to the predicate that describes what they are known for [20]. Link prediction on KGs refers to identifying such missing information [9]. Knowledge Graph Embedding (KGE) models have been particularly successful at tackling the link prediction task [24].

We investigate the use of a 2D convolution in the complex space $\mathbb{C}$ to tackle the link prediction task. We are especially interested in an effective composition

---

[1] `github.com/dice-group/Convolutional-Complex-Knowledge-Graph-Embeddings`

of the non-symmetric property of Hermitian products with the parameter sharing property of a 2D convolution. Previously, Trouillon et al. [35] showed the expressiveness of a Hermitian product on complex-valued embeddings $\text{Re}(\langle \mathbf{e}_h, \mathbf{e}_r, \overline{\mathbf{e}_t} \rangle)$, where $\mathbf{e}_h$, $\mathbf{e}_r$, and $\mathbf{e}_t$ stand for the embeddings of head entity, relation and tail entity, respectively; $\overline{\mathbf{e}_t}$ is the complex conjugate of $\mathbf{e}_t$. The Hermitian product used in [35] is not symmetric and can be used to model antisymmetric relations since $\text{Re}(\langle \mathbf{e}_h, \mathbf{e}_r, \overline{\mathbf{e}_t} \rangle) \neq \text{Re}(\langle \mathbf{e}_t, \mathbf{e}_r, \overline{\mathbf{e}_h} \rangle)$. Dettmers et al. [9] and Nguyen et al. [23] indicated the effectiveness of using a 2D convolution followed by an affine transformation to predict missing links. Additionally, Balažević et al. [3] showed that 1D relation-specific convolution filters can be an effective means to tackle the link prediction task. Chen et al. [6] suggested applying a 2D convolution followed by two capsule layers on quaternion-valued embeddings. In turn, the results of a recent work [28] highlighted the importance of extensive hyperparameter optimization and new training strategies (see Table 1). The paper showed that the link prediction performances of previous state-of-the-art models (e.g., RESCAL, ComplEx and DistMult [26,35,37]) increased by up to 10% absolute on benchmark datasets, provided that new training strategies are applied. Based on these considerations, we propose CONEX—a multiplicative composition of a 2D convolution operation with a Hermitian inner product of complex-valued embedding vectors. By virtue of its novel architecture, CONEX is able to control the impact of a 2D convolution on predicted scores, i.e., by endowing ComplEx with two more degrees of freedom (see Section 4). Ergo, CONEX is able to degenerate to ComplEx if such a degeneration is necessary to further reduce the incurred training loss.

We evaluated CONEX on five of the most commonly used benchmark datasets (WN18, WN18RR, FB15K, FB15K-237 and YAGO3-10). We used the findings of [28] on using Bayesian optimization to select a small sample of hyperparameter values for our experiments. Hence, we did not need to perform an extensive hyperparameter optimization throughout our experiments and fixed the seed for the pseudo-random generator to 1. In our experiments, we followed the standard training strategy commonly used in the literature [4,3]. Overall, our results suggest that CONEX outperforms state-of-the-art models on four out of five benchmark datasets w.r.t. Hits@N and Mean Reciprocal Rank (MRR). CONEX outperforms ComplEx and ConvE on all benchmark datasets in all metrics. Results of our statistical hypothesis testing indicates that the superior performance of CONEX is statistically significant. Our ablation study suggests that the dropout technique and the label smoothing have the highest impact on the performance of CONEX. Furthermore, our results on the YAGO3-10 dataset supports the findings of Ruffinelli et al. [28] as training DistMult and ComplEx with new techniques resulted in increasing their MRR performances by absolute 20% and 19%, respectively. Finally, our results suggest that the generalization performance of models can be further improved by applying ensemble learning. In particular, ensembling CONEX leads to a new state-of-the-art performance on WN18RR and FB15K-237.

## 2   Related work

A wide range of works have investigated KGE to address various tasks such as type prediction, relation prediction, link prediction, question answering, item recommendation and knowledge graph completion [8,7,26,14]. We refer to [24,36,5,16,27] for recent surveys and give a brief overview of selected KGE techniques. Table 1 shows scoring functions of state-of-the-art KGE models.

RESCAL [26] is a bilinear model that computes a three-way factorization of a third-order adjacency tensor representing the input KG. RESCAL captures various types of relations in the input KG but is limited in its scalability as it has quadratic complexity in the factorization rank [33]. DistMult [37] can be seen as an efficient extension of RESCAL with a diagonal matrix per relation to reduce the complexity of RESCAL [4]. DistMult performs poorly on antisymmetric relations while performing well on symmetric relations [33]. Note that through applying the reciprocal data augmentation technique, this incapability of DistMult is alleviated [28]. TuckER [4] performs a Tucker decomposition on the binary tensor representing the input KG, which enables multi-task learning through parameter sharing between different relations via the core tensor.

Table 1: State-of-the-art KGE models with training strategies. $\mathbf{e}$ denotes embeddings, $\overline{\mathbf{e}} \in \mathbb{C}$ corresponds to the complex conjugate of $\mathbf{e}$. $*$ denotes a convolution operation with $\omega$ kernel. $f$ denotes rectified linear unit function. $\otimes, \circ, \cdot$ denote the Hamilton, the Hadamard and an inner product, respectively. In ConvE, the reshaping operation is omitted. The tensor product along the $n$-th mode is denoted by $\times_n$ and the core tensor is represented by $\mathcal{W}$. MSE, MR, BCE and CE denote mean squared error, margin ranking, binary cross entropy and cross entropy loss functions. NegSamp and AdvNegSamp stand for negative sampling and adversarial sampling.

| Model | Scoring Function | VectorSpace | Loss | Training | Optimizer | Regularizer |
|---|---|---|---|---|---|---|
| RESCAL [26] | $\mathbf{e}_h \cdot \mathcal{W}_r \cdot \mathbf{e}_t$ | $\mathbf{e}_h, \mathbf{e}_t \in \mathbb{R}$ | MSE | Full | ALS | L2 |
| DistMult [37] | $\langle \mathbf{e}_h, \mathbf{e}_r, \mathbf{e}_t \rangle$ | $\mathbf{e}_h, \mathbf{e}_r, \mathbf{e}_t \in \mathbb{R}$ | MR | NegSamp | Adagrad | Weighted L2 |
| ComplEx [35] | $\text{Re}(\langle \mathbf{e}_h, \mathbf{e}_r, \overline{\mathbf{e}_t} \rangle)$ | $\mathbf{e}_h, \mathbf{e}_r, \mathbf{e}_t \in \mathbb{C}$ | BCE | NegSamp | Adagrad | Weighted L2 |
| ConvE [9] | $f(\text{vec}(f([\mathbf{e}_h; \mathbf{e}_r] * \omega))\mathbf{W}) \cdot \mathbf{e}_t$ | $\mathbf{e}_h, \mathbf{e}_r, \mathbf{e}_t \in \mathbb{R}$ | BCE | KvsAll | Adam | Dropout, BatchNorm |
| TuckER [4] | $\mathcal{W} \times_1 \mathbf{e}_h \times_2 \mathbf{e}_r \times_3 \mathbf{e}_t$ | $\mathbf{e}_h, \mathbf{e}_r, \mathbf{e}_t \in \mathbb{R}$ | BCE | KvsAll | Adam | Dropout, BatchNorm |
| RotatE [31] | $- \parallel \mathbf{e}_h \circ \mathbf{e}_r - \mathbf{e}_t \parallel$ | $\mathbf{e}_h, \mathbf{e}_r, \mathbf{e}_t \in \mathbb{C}$ | CE | AdvNegSamp | Adam | - |
| QuatE [38] | $\mathbf{e}_h \otimes \mathbf{e}_r^{\triangleleft} \cdot \mathbf{e}_t$ | $\mathbf{e}_h, \mathbf{e}_r, \mathbf{e}_t \in \mathbb{H}$ | CE | AdvNegSamp | Adagrad | Weighted L2 |
| CONEX | $\text{conv}(\mathbf{e}_h, \mathbf{e}_r) \circ \text{Re}(\langle \mathbf{e}_h, \mathbf{e}_r, \overline{\mathbf{e}_t} \rangle)$ | $\mathbf{e}_h, \mathbf{e}_r, \mathbf{e}_t \in \mathbb{C}$ | BCE | KvsAll | Adam | Dropout, BatchNorm |

ComplEx [35] extends DistMult by learning representations in a complex vector space. ComplEx is able to infer both symmetric and antisymmetric relations via a Hermitian inner product of embeddings that involves the conjugate-transpose of one of the two input vectors. ComplEx yields state-of-the-art performance on the link prediction task while leveraging linear space and time complexity of the dot products. Trouillon et. al. [34] showed that ComplEx is equivalent to HolE [25]. Inspired by Euler's identity, RotatE [31] employs a rotational model taking

predicates as rotations from subjects to objects in complex space via the element-wise Hadamard product [16]. RotatE performs well on composition/transitive relations while ComplEx performs poorly [31]. QuatE [38] extends the complex-valued space into hypercomplex by a quaternion with three imaginary components, where the Hamilton product is used as compositional operator for hypercomplex valued-representations.

ConvE [9] applies a 2D convolution to model the interactions between entities and relations. Through interactions captured by 2D convolution, ConvE yields a state-of-art performance in link prediction. ConvKB extends ConvE by omitting the reshaping operation in the encoding of representations in the convolution operation [23]. Similarly, HypER extends ConvE by applying relation-specific convolution filters as opposed to applying filters from concatenated subject and relation vectors [3].

## 3    Preliminaries and Notation

### 3.1    Knowledge Graphs

Let $\mathcal{E}$ and $\mathcal{R}$ represent the set of entities and relations, respectively. Then, a KG $\mathcal{G} = \{(\mathbf{h}, \mathbf{r}, \mathbf{t}) \in \mathcal{E} \times \mathcal{R} \times \mathcal{E}\}$ can be formalised as a set of triples where each triple contains two entities $\mathbf{h}, \mathbf{t} \in \mathcal{E}$ and a relation $\mathbf{r} \in \mathcal{R}$. A relation $\mathbf{r}$ in $\mathcal{G}$ is

- *symmetric* if $(\mathbf{h}, \mathbf{r}, \mathbf{t}) \iff (\mathbf{t}, \mathbf{r}, \mathbf{h})$ for all pairs of entities $\mathbf{h}, \mathbf{t} \in \mathcal{E}$,
- *anti-symmetric* if $(\mathbf{h}, \mathbf{r}, \mathbf{t}) \in \mathcal{G} \Rightarrow (\mathbf{t}, \mathbf{r}, \mathbf{h}) \notin \mathcal{G}$ for all $\mathbf{h} \neq \mathbf{t}$, and
- *transitive/composite* if $(\mathbf{h}, \mathbf{r}, \mathbf{t}) \in \mathcal{G} \wedge (\mathbf{t}, \mathbf{r}, \mathbf{y}) \in \mathcal{G} \Rightarrow (\mathbf{h}, \mathbf{r}, \mathbf{y}) \in \mathcal{G}$ for all $\mathbf{h}, \mathbf{t}, \mathbf{y} \in \mathcal{E}$ [31,18].

The inverse of a relation $\mathbf{r}$, denoted $\mathbf{r}^{-1}$, is a relation such that for any two entities $\mathbf{h}$ and $\mathbf{t}$, $(\mathbf{h}, \mathbf{r}, \mathbf{t}) \in \mathcal{G} \iff (\mathbf{t}, \mathbf{r}^{-1}, \mathbf{h}) \in \mathcal{G}$.

### 3.2    Link Prediction

The link prediction refers to predicting whether unseen triples (i.e., triples not found in $\mathcal{G}$) are true [16]. The task is often formalised by learning a scoring function $\phi : \mathcal{E} \times \mathcal{R} \times \mathcal{E} \mapsto \mathbb{R}$ [24,16] ideally characterized by $\phi(\mathbf{h}, \mathbf{r}, \mathbf{t}) > \phi(\mathbf{x}, \mathbf{y}, \mathbf{z})$ if $(\mathbf{h}, \mathbf{r}, \mathbf{t})$ is true and $(\mathbf{x}, \mathbf{y}, \mathbf{z})$ is not.

## 4    Convolutional Complex Knowledge Graph Embeddings

Inspired by the previous works ComplEx [35] and ConvE [9], we dub our approach CONEX (convolutional complex knowledge graph embeddings).

**Motivation.** Sun et. al. [31] suggested that ComplEx is not able to model triples with transitive relations since ComplEx does not perform well on datasets containing many transitive relations (see Table 5 and Section 4.6 in [31]). Motivated by this consideration, we propose CONEX, which applies the Hadamard product to compose a 2D convolution followed by an affine transformation with a Hermitian inner product in $\mathbb{C}$. By virtue of the proposed architecture (see Equation (1)), CONEX is endowed with the capability of

1. leveraging a 2D convolution and
2. degenerating to ComplEx if such degeneration is necessary to further minimize the incurred training loss.

CONEX benefits from the *parameter sharing* and *equivariant representation* properties of convolutions [11]. The parameter sharing property of the convolution operation allows CONEX to achieve parameter efficiency, while the equivariant representation allows CONEX to effectively integrate interactions captured in the stacked complex-valued embeddings of entities and relations into computation of scores. This implies that small interactions in the embeddings have small impacts on the predicted scores[2]. The rationale behind this architecture is to increase the expressiveness of our model without increasing the number of its parameters. As previously stated in [35], this nontrivial endeavour is the keystone of embedding models. Ergo, we aim to overcome the shortcomings of ComplEx in modelling triples containing transitive relations through combining it with a 2D convolutions followed by an affine transformation on $\mathbb{C}$.

**Approach.** Given a triple $(\mathtt{h}, \mathtt{r}, \mathtt{t})$, $\text{CONEX} : \mathbb{C}^{3d} \mapsto \mathbb{R}$ computes its score as

$$\text{CONEX}(\mathtt{h}, \mathtt{r}, \mathtt{t}) = \text{conv}(\mathbf{e}_h, \mathbf{e}_r) \circ \text{Re}(\langle \mathbf{e}_h, \mathbf{e}_r, \overline{\mathbf{e}_t} \rangle), \tag{1}$$

where $\text{conv}(\cdot, \cdot) : \mathbb{C}^{2d} \mapsto \mathbb{C}^d$ is defined as

$$\text{conv}(\mathbf{e}_h, \mathbf{e}_r) = f\big( \text{vec}(f([\mathbf{e}_h, \mathbf{e}_r] * \omega)) \cdot \mathbf{W} + \mathbf{b} \big), \tag{2}$$

where $f(\cdot)$ denotes the rectified linear unit function (ReLU), $\text{vec}(\cdot)$ stands for a flattening operation, $*$ is the convolution operation, $\omega$ stands for kernels/filters in the convolution, and $(\mathbf{W}, \mathbf{b})$ characterize an affine transformation. By virtue of its novel structure, CONEX is enriched with the capability of controlling the impact of a 2D convolution and Hermitian inner product on the predicted scores. Ergo, the gradients of loss (see Equation (6)) w.r.t. embeddings can be propagated in two ways, namely, via $\text{conv}(\mathbf{e}_h, \mathbf{e}_r)$ or $\text{Re}(\langle \mathbf{e}_h, \mathbf{e}_r, \overline{\mathbf{e}_t} \rangle)$. Equation (1) can be

---

[2] We refer to [11] for further details of properties of convolutions.

equivalently expressed by expanding its real and imaginary parts:

$$\text{ConEx}(h, r, t) = \sum_{k=1}^{d} \text{Re}(\gamma)_k \text{Re}(\mathbf{e}_h)_k \text{Re}(\mathbf{e}_r)_k \text{Re}(\overline{\mathbf{e}_t})_k \tag{3}$$

$$= \langle \text{Re}(\gamma), \text{Re}(\mathbf{e}_h), \text{Re}(\mathbf{e}_r), \text{Re}(\mathbf{e}_t) \rangle$$
$$+ \langle \text{Re}(\gamma), \text{Re}(\mathbf{e}_h), \text{Im}(\mathbf{e}_r), \text{Im}(\mathbf{e}_t) \rangle$$
$$+ \langle \text{Im}(\gamma), \text{Im}(\mathbf{e}_h), \text{Re}(\mathbf{e}_r), \text{Im}(\mathbf{e}_t) \rangle$$
$$- \langle \text{Im}(\gamma), \text{Im}(\mathbf{e}_h), \text{Im}(\mathbf{e}_r), \text{Re}(\mathbf{e}_t) \rangle \tag{4}$$

where $\overline{\mathbf{e}_t}$ is the conjugate of $\mathbf{e}_t$ and $\gamma$ denotes the output of $\text{conv}(\mathbf{e}_h, \mathbf{e}_r)$ for brevity. Such multiplicative inclusion of $\text{conv}(\cdot, \cdot)$ equips ConEx with two more degrees of freedom due the $\text{Re}(\gamma)$ and $\text{Im}(\gamma)$ parts.

**Connection to ComplEx.** During the optimization, $\text{conv}(\cdot, \cdot)$ is allowed to reduce its range into $\gamma \in \mathbb{C}$ such that $\text{Re}(\gamma) = 1 \wedge \text{Im}(\gamma) = 1$ . This allows ConEx to degenerate into ComplEx as shown in Equation (5):

$$\text{ConEx}(\mathbf{h}, \mathbf{r}, \mathbf{t}) = \gamma \circ \text{ComplEx}(\mathbf{h}, \mathbf{r}, \mathbf{t}). \tag{5}$$

This multiplicative inclusion of $\text{conv}(\cdot, \cdot)$ is motivated by the scaling parameter in the batch normalization (see section 3 in [15]). Consequently, ConEx is allowed use a 2D convolution followed by an affine transformation as a scaling factor in the computation of scores.

**Training.** We train our approach by following a standard setting [9,4]. Similarly, we applied the standard data augmentation technique, the KvsAll training procedure[3]. After the data augmentation technique for a given pair $(\mathbf{h}, \mathbf{r})$, we compute scores for all $\mathbf{x} \in \mathcal{E}$ with $\phi(\mathbf{h}, \mathbf{r}, \mathbf{x})$. We then apply the logistic sigmoid function $\sigma(\phi((\mathbf{h}, \mathbf{r}, \mathbf{t})))$ to obtain predicted probabilities of entities. ConEx is trained to minimize the binary cross entropy loss function $L$ that determines the incurred loss on a given pair $(\mathbf{h}, \mathbf{r})$ as defined in the following:

$$L = -\frac{1}{|\mathcal{E}|} \sum_{i=1}^{|\mathcal{E}|} (\mathbf{y}^{(i)} \log(\hat{\mathbf{y}}^{(i)}) + (1 - \mathbf{y}^{(i)}) \log(1 - \hat{\mathbf{y}}^{(i)})), \tag{6}$$

where $\hat{\mathbf{y}} \in \mathbb{R}^{|\mathcal{E}|}$ is the vector of predicted probabilities and $\mathbf{y} \in [0, 1]^{|\mathcal{E}|}$ is the binary label vector.

## 5   Experiments

### 5.1   Datasets

We used five of the most commonly used benchmark datasets (WN18, WN18RR, FB15K, FB15K-237 and YAGO3-10). An overview of the datasets is provided

---

[3] Note that the KvsAll strategy is called 1-N scoring in [9]. Here, we follow the terminology of [28].

in Table 2. WN18 and WN18RR are subsets of Wordnet, which describes lexical and semantic hierarchies between concepts and involves **symmetric** and **anti-symmetric** relation types, while FB15K, FB15K-237 are subsets of Freebase, which involves mainly **symmetric**, **antisymmetric** and **composite** relation types [31]. We refer to [9] for further details pertaining to the benchmark datasets.

Table 2: Overview of datasets in terms of number of entities, number of relations, and node degrees in the train split along with the number of triples in each split of the dataset.

| **Dataset** | $|\mathcal{E}|$ | $|\mathcal{R}|$ | Degr. (M±SD) | $|\mathcal{G}^{\text{Train}}|$ | $|\mathcal{G}^{\text{Validation}}|$ | $|\mathcal{G}^{\text{Test}}|$ |
|---|---|---|---|---|---|---|
| YAGO3-10 | 123,182 | 37 | 9.6±8.7 | 1,079,040 | 5,000 | 5,000 |
| FB15K | 14,951 | 1,345 | 32.46±69.46 | 483,142 | 50,000 | 59,071 |
| WN18 | 40,943 | 18 | 3.49±7.74 | 141,442 | 5,000 | 5,000 |
| FB15K-237 | 14,541 | 237 | 19.7±30 | 272,115 | 17,535 | 20,466 |
| WN18RR | 40,943 | 11 | 2.2±3.6 | 86,835 | 3,034 | 3,134 |

### 5.2   Evaluation Metrics

We used the filtered MRR and Hits@N to evaluate link prediction performances, as in previous works [31,35,9,4]. We refer to [28] for details pertaining to metrics.

### 5.3   Experimental Setup

We selected the hyperparameters of CONEX based on the MRR score obtained on the validation set of WN18RR. Hence, we evaluated the link prediction performance of CONEX on FB15K-237, YAGO3-10, WN18 and FB15K by using the best hyperparameter configuration found on WN18RR. This decision stems from the fact that we aim to reduce the impact of extensive hyperparameter optimization on the reported results and the $CO_2$ emission caused through relying on the findings of previously works [28]. Strubell et al. [30] highlighted the substantial energy consumption of performing extensive hyperparameter optimization. Moreover, Ruffinelli et al. [28] showed that model configurations can be found by exploring relatively few random samples from a large hyperparameter space. With these considerations, we determined the ranges of hyperparameters for the grid search algorithm optimizer based on their best hyperparameter setting for ConvE (see Table 8 in [28]). Specifically, the ranges of the hyperparameters were defined as follows: $d$: $\{100, 200\}$; dropout rate:$\{.3, .4\}$ for the input; dropout rate: $\{.4, .5\}$ for the feature map; label smoothing: $\{.1\}$ and the number of output channels in the convolution operation: $\{16, 32\}$; the batch size: $\{1024\}$; the learning rate: $\{.001\}$. After determining the best hyperparameters based on the

MRR on the validation dataset; we retrained CONEX with these hyperparameters on the combination of train and valid sets as applied in [17].

Motivated by the experimental setups for ResNet [12] and AlexNet [19], we were interested in quantifying the impact of ensemble learning on the link prediction performances. Ensemble learning refers to learning a weighted combination of learning algorithms. In our case, we generated ensembles of models by averaging the predictions of said models.[4] To this end, we re-evaluated state-of-the-art models, including TucKER, DistMult and ComplEx on the combination of train and validation sets of benchmark datasets. Therewith, we were also able to quantify the impact of training state-of-the-art models on the combination of train and validation sets. Moreover, we noticed that link prediction performances of DistMult and ComplEx, on the YAGO3-10 dataset were reported without employing new training strategies (KvsAll, the reciprocal data augmentation, the batch normalization, and the ADAM optimizer). Hence, we trained DistMult, ComplEx on YAGO3-10 with these strategies.

### 5.4   Implementation Details and Reproducibility

We implemented and evaluated our approach in the framework provided by [4,2]. Throughout our experiments, the seed for the pseudo-random generator was fixed to 1. To alleviate the hardware requirements for the reproducibility of our results and to foster further reproducible research, we provide hyperparameter optimization, training and evaluation scripts along with pretrained models at the project page.

## 6   Results

Table 3, Table 4 and Table 10 report the link prediction performances of CONEX on five benchmark datasets. Overall, CONEX outperforms state-of-the-art models on four out of five datasets. In particular, CONEX outperforms ComplEx and ConvE on all five datasets. This supports our original hypothesis, i.e., that the composition of a 2D convolution with a Hermitian inner product improves the prediction of relations in complex spaces. We used the Wilcoxon signed-rank test to measure the statistical significance of our link prediction results. Moreover, we performed an ablation study (see Table 8) to obtain confidence intervals for prediction performances of CONEX. These results are shown in the Appendix. Bold and underlined entries denote best and second-best results in all tables.

CONEX outperforms all state-of-the-art models on WN18 and FB15K (see Table 10 in the Appendix), whereas such distinct superiority is not observed on WN18RR and FB15K-237. Table 3 shows that CONEX outperforms many state-of-the-art models, including RotatE, ConvE, HypER, ComplEx, NKGE, in all metrics on WN18RR and FB15K-237. This is an important result for two reasons:

---

[4] Ergo, the weights for models were set to 1 (see the section 16.6 in [22] for more details.)

Table 3: Link prediction results on WN18RR and FB15K-237. Results are obtained from corresponding papers. ‡ represents recently reported results of corresponding models.

| | WN18RR | | | | FB15K-237 | | | |
|---|---|---|---|---|---|---|---|---|
| | MRR | Hits@10 | Hits@3 | Hits@1 | MRR | Hits@10 | Hits@3 | Hits@1 |
| DistMult [9] | .430 | .490 | .440 | .390 | .241 | .419 | .263 | .155 |
| ComplEx [9] | .440 | .510 | .460 | .410 | .247 | .428 | .275 | .158 |
| ConvE [9] | .430 | .520 | .440 | .400 | .335 | .501 | .356 | .237 |
| RESCAL$^{\dagger}$ [28] | .467 | .517 | .480 | .439 | .357 | .541 | .393 | .263 |
| DistMult$^{\dagger}$ [28] | .452 | .530 | .466 | .413 | .343 | .531 | .378 | .250 |
| ComplEx$^{\dagger}$ [28] | .475 | .547 | .490 | .438 | .348 | .536 | .384 | .253 |
| ConvE$^{\dagger}$ [28] | .442 | .504 | .451 | .411 | .339 | .521 | .369 | .248 |
| HypER [3] | .465 | .522 | .477 | .436 | .341 | .520 | .376 | .252 |
| NKGE [38] | .450 | .526 | .465 | .421 | .330 | .510 | .365 | .241 |
| RotatE [31] | .476 | .571 | .492 | .428 | .338 | .533 | .375 | .241 |
| TuckER [4] | .470 | .526 | .482 | .443 | .358 | .544 | .394 | .266 |
| QuatE [38] | **.482** | **.572** | **.499** | .436 | **.366** | **.556** | .401 | **.271** |
| DistMult | .439 | .527 | .455 | .399 | .353 | .539 | .390 | .260 |
| ComplEx | .453 | .546 | .473 | .408 | .332 | .509 | .366 | .244 |
| TuckER | .466 | .515 | .476 | .441 | .363 | .553 | .400 | .268 |
| ConEx | .481 | .550 | .493 | **.448** | **.366** | .555 | **.403** | **.271** |

(1) ConEx requires significantly fewer parameters to yield such superior results (e.g., ConEx only requires 26.63M parameters on WN18RR, while RotatE relies on 40.95M parameters), and (2) we did not tune the hyperparameters of ConEx on FB15K-237.Furthermore, the results reported in Table 3 corroborate the findings of Ruffinelli et al. [28]: training DistMult and ComplEx with KvsAll, the reciprocal data augmentation, the batch normalization, and the ADAM optimizer leads to a significant improvement, particularly on FB15K-237.

During our experiments, we observed that many state-of-the-art models are not evaluated on YAGO3-10. This may stem from the fact that the size of YAGO3-10 prohibits performing extensive hyperparameter optimization even with the current state-of-the-art hardware systems. Note that YAGO3-10 involves 8.23 and 8.47 times more entities than FB15K and FB15K-237, respectively. Table 4 indicates that DistMult and ComplEx perform particularly well on YAGO3-10, provided that KvsAll, the reciprocal data augmentation, the batch normalization, and the ADAM optimizer are employed. These results support findings of Ruffinelli et al. [28]. During training, we observed that the training loss of DistMult and ComplEx seemed to converge within 400 epochs, whereas the training loss of TuckER seemed to continue decreasing. Ergo, we conjecture that TuckER is more likely to benefit from increasing the number of epochs than DistMult and ComplEx. Table 4 shows that the superior performance of ConEx against state-

Table 4: Link prediction results on YAGO3-10. Results are obtained from corresponding papers.

|  | YAGO3-10 | | | |
|---|---|---|---|---|
|  | MRR | Hits@10 | Hits@3 | Hits@1 |
| DistMult [9] | .340 | .540 | .380 | .240 |
| ComplEx [9] | .360 | .550 | .400 | .260 |
| ConvE [9] | .440 | .620 | .490 | .350 |
| HypER [3] | .533 | .678 | .580 | .455 |
| RotatE [31] | .495 | .670 | .550 | .402 |
| DistMult | .543 | .683 | .590 | .466 |
| ComplEx | .547 | .690 | .594 | .468 |
| TuckER | .427 | .609 | .476 | .331 |
| CONEX | **.553** | **.696** | **.601** | **.474** |

of-the-art models including DistMult, ComplEx, HypER can be maintained on the largest benchmark dataset for the link prediction.

Delving into the link prediction results, we observed an inconsistency in the test splits of WN18RR and FB15K-237. Specifically, the test splits of WN18RR and FB15K-237 contain many out-of-vocabulary entities[5]. For instance, 6% of the test set on WN18RR involves out-of-vocabulary entities. During our experiments, we did not remove such triples to obtain fair comparisons on both datasets. To quantify the impact of unseen entities on link prediction performances, we conducted an additional experiment.

**Link Prediction per Relation.** Table 5 reports the link prediction per relation performances on WN18RR. Overall, models perform particularly well on triples containing symmetric relations such as `also_see` and `similar_to`. Compared to RotatE, DistMult, ComplEx and TuckER, CONEX performs well on triples containing transitive relations such as `hypernym` and `has_part`. Allen et al. [1] ranked the complexity of type of relations as R > S > C in the link prediction task. Based on this ranking, superior performance of CONEX becomes more apparent as the complexity of relations increases.

**Ensemble Learning.** Table 6 reports the link prediction performances of ensembles based on pairs of models. These results suggest that ensemble learning can be applied as an effective means to boost the generalization performance of existing approaches including CONEX. These results may also indicate that models may be further improved through optimizing the impact of each model on the ensembles, e.g., by learning two scalars $\alpha$ and $\beta$ in $\big(\alpha\text{CONEX}(\mathbf{s},\mathbf{p},\mathbf{o}) + \beta\text{TuckER}(\mathbf{s},\mathbf{p},\mathbf{o})\big)$ instead of averaging predicted scores.

---

[5] `github.com/TimDettmers/ConvE/issues/66`

Table 5: MRR link prediction on each relation of WN18RR. Results of RotatE are taken from [38]. The complexity of type of relations in the link prediction task is defined as R > S > C [1].

| Relation Name | Type | RotatE | DistMult | ComplEx | TuckER | ConEx |
|---|---|---|---|---|---|---|
| hypernym | S | .148 | .102 | .106 | .121 | **.149** |
| instance_hypernym | S | .318 | .218 | .292 | .375 | **.393** |
| member_meronym | C | **.232** | .129 | .181 | .181 | .171 |
| synset_domain_topic_of | C | .341 | .226 | .266 | .344 | **.373** |
| has_part | C | .184 | .143 | .181 | .171 | **.192** |
| member_of_domain_usage | C | **.318** | .225 | .280 | .213 | **.318** |
| member_of_domain_region | C | .200 | .095 | .267 | .284 | **.354** |
| derivationally_related_form | R | .947 | .982 | .984 | .985 | **.986** |
| also_see | R | .585 | .639 | .557 | **.658** | .647 |
| verb_group | R | .943 | **1.00** | **1.00** | **1.00** | **1.00** |
| similar_to | R | **1.00** | **1.00** | **1.00** | **1.00** | **1.00** |

Table 6: Link prediction results of ensembled models on WN18RR and FB15K-237. Second rows denote link prediction results without triples containing out-of-vocabulary entities. CONEX-CONEX stands for ensembling two CONEX trained with the dropout rate 0.4 and 0.5 on the feature map.

| | WN18RR | | | | FB15K-237 | | | |
|---|---|---|---|---|---|---|---|---|
| | MRR | Hits@10 | Hits@3 | Hits@1 | MRR | Hits@10 | Hits@3 | Hits@1 |
| DistMult-ComplEx | .446 | .545 | .467 | .398 | .359 | .546 | .397 | .265 |
| | .475 | .579 | .497 | .426 | .359 | .546 | .397 | .265 |
| DistMult-TuckER | .446 | .533 | .461 | .405 | .371 | .563 | .410 | .275 |
| | .476 | .569 | .492 | .433 | .371 | .563 | .411 | .275 |
| CONEX-DistMult | .454 | .545 | .471 | .410 | .371 | .563 | .409 | .275 |
| | .484 | .580 | .501 | .439 | .367 | .556 | .403 | .272 |
| CONEX-ComplEx | .470 | .554 | .487 | .428 | .370 | .559 | .407 | .276 |
| | .501 | .589 | .518 | .456 | .360 | .547 | .397 | .267 |
| CONEX-TuckER | .483 | .549 | .494 | .449 | .375 | .568 | .414 | .278 |
| | .514 | .583 | **.526** | **.479** | .375 | .568 | .414 | .278 |
| CONEX-CONEX | .485 | .559 | .495 | .450 | **.376** | .569 | **.415** | **.279** |
| | **.517** | **.594** | **.526** | **.479** | **.376** | **.570** | **.415** | **.279** |

**Parameter Analysis.** Table 7 indicates the robustness of CONEX against the overfitting problem. Increasing the number of parameters in CONEX does not lead to a significant decrease in the generalization performance. In particular, CONEX achieves similar generalization performance, with $p = 26.63M$ and $p = 70.66M$,

Table 7: Influence of different hyperparameter configurations for CONEX on WN18RR. $d$, $c$ and $p$ stand for the dimensions of embeddings in $\mathbb{C}$, number of output channels in 2D convolutions and number of free parameters in millions, respectively.

| | | | WN18RR | | | |
|---|---|---|---|---|---|---|
| $d$ | $c$ | $p$ | MRR | Hits@10 | Hits@3 | Hits@1 |
| 300 | 64 | 70.66M | .475 | .540 | .490 | .442 |
| 250 | 64 | 52.49M | .475 | .541 | .488 | .441 |
| 300 | 32 | 47.62M | .480 | .548 | .491 | .447 |
| 250 | 32 | 36.39M | .479 | .545 | .490 | .446 |
| 300 | 16 | 36.10M | .479 | .550 | .494 | .445 |
| 250 | 16 | 28.48M | .477 | .544 | .489 | .443 |
| 200 | 32 | 26.63M | .481 | .550 | .493 | .447 |
| 100 | 32 | 10.75M | .474 | .533 | .480 | .440 |
| 100 | 16 | 9.47M | .476 | .536 | .486 | .441 |
| 50 | 32 | 4.74M | .448 | .530 | .477 | .401 |

as the difference between MRR scores are less than absolute 1%. This cannot be explained with convolutions playing no role as CONEX would then degrade back to ComplEx and achieve the same results (which is clearly not the case in our experiments).

## 7   Discussion

The superior performance of CONEX stems from the composition of a 2D convolution with a Hermitian inner product of complex-valued embeddings. Trouillon et al. [35] showed that a Hermitian inner product of complex-valued embeddings can be effectively used to tackle the link prediction problem. Applying the convolution operation on complex-valued embeddings of subjects and predicates permits CONEX to recognize interactions between subjects and predicates in the form of complex-valued feature maps. Through the affine transformation of feature maps and their inclusion into a Hermitian inner product involving the conjugate-transpose of complex-valued embeddings of objects, CONEX can accurately infer various types of relations. Moreover, the number and shapes of the kernels permit to adjust the expressiveness , while CONEX retains the parameter efficiency due to the parameter sharing property of convolutions. By virtue of the design, the expressiveness of CONEX may be further improved by increasing the depth of the conv$(\cdot, \cdot)$ via the residual learning block [12].

## 8   Conclusion and Future Work

In this work, we introduced CONEX—a multiplicative composition of a 2D convolution with a Hermitian inner product on complex-valued embeddings. By

virtue of its novel structure, CONEX is endowed with the capability of controlling the impact of a 2D convolution and a Hermitian inner product on the predicted scores. Such combination makes CONEX more robust to overfitting, as is affirmed with our parameter analysis. Our results open a plethora of other research avenues. In future work, we plan to investigate the following: (1) combining the convolution operation with Hypercomplex multiplications, (2) increasing the depth in the convolutions via residual learning block and (3) finding more effective combinations of ensembles of models.

## Acknowledgments

This work has been supported by the BMWi-funded project RAKI (01MD19012D) as well as the BMBF-funded project DAIKIRI (01IS19085B). We are grateful to Diego Moussallem for valuable comments on earlier drafts and to Pamela Heidi Douglas for editing the manuscript.

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

## 9     Appendix

**Statistical Hypothesis Testing.** We carried out a Wilcoxon signed-rank test to check whether our results are significant. Our null hypothesis was that the link prediction performances of CONEX, ComplEx and ConvE come from the same distribution. The alternative hypothesis was correspondingly that these results come from different distributions. To perform the Wilcoxon signed-rank test (two-sided), we used the differences of the MRR, Hits@1, Hits@3, and Hits@10 performances on WN18RR, FB15K-237 and YAGO3-10. We performed two hypothesis tests between CONEX and ComplEx as well as between CONEX and ConvE. In both tests, we were able to reject the null hypothesis with a p-value $< 1\%$. Ergo, the superior performance of CONEX is statistically significant.

**Ablation Study.** We conducted our ablation study in a fashion akin to [9]. Like [9], we evaluated 2 different parameter initialisations to compute confidence intervals that is defined as $\bar{x} \pm 1.96 \cdot \frac{s}{\sqrt{n}}$, where $\bar{x} = \frac{1}{n} \sum_i^n x_i$ and $s = \sqrt{\frac{\sum_i^n (x_i - \bar{x})^2}{n}}$, respectively. Hence, the mean and the standard deviation are computed without Bessel's correction. Our results suggest that the initialization of parameters does not play a significant role in the link performance of CONEX. The dropout technique is the most important component in the generalization performance of CONEX. This is also observed in [9]. Moreover, replacing the Adam optimizer with the RMSprop optimizer [32] leads to slight increases in the variance of the link prediction results. During our ablation experiments, we were also interested in decomposing CONEX through removing $\mathrm{conv}(\cdot, \cdot)$, after CONEX is trained with it on benchmark datasets. By doing so, we aim to observe the impact of a 2D convolution in the computation of scores. Table 9 indicates that the impact of $\mathrm{conv}(\cdot, \cdot)$ differs depending on the input knowledge graph. As the size of the input knowledge graph increases, the impact of $\mathrm{conv}(\cdot, \cdot)$ on the computation of scores of triples increases.

Table 8: Ablation study for CONEX on FB15K-237. *dp* and *ls* denote the dropout technique and the label smoothing technique, respectively.

|  | FB15K-237 | | | |
|---|---|---|---|---|
|  | MRR | Hits@10 | Hits@3 | Hits@1 |
| Full | .366±.000 | .556±.001 | .404±.001 | .270±.001 |
| No *dp* on inputs | .282±.000 | .441±.001 | .313±.001 | .203±.000 |
| No *dp* on feature map | .351±.000 | .533±.000 | .388±.001 | .259±.001 |
| No *ls* | .321±.001 | .498±.001 | .354±.001 | .232±.002 |
| With RMSprop | .361±.004 | .550±.007 | .400±.005 | .267±.003 |

Table 9: Link prediction results on benchmark datasets. CONEX⁻ stands for removing conv(·, ·) in CONEX during the evaluation.

|  | ConEx | | | | ConEx⁻ | | | |
|---|---|---|---|---|---|---|---|---|
|  | MRR | Hits@10 | Hits@3 | Hits@1 | MRR | Hits@10 | Hits@3 | Hits@1 |
| **WN18RR** | .481 | .550 | .493 | .448 | .401 | .494 | .437 | .346 |
| **FB15K-237** | .366 | .555 | .403 | .271 | .284 | .458 | .314 | .198 |
| **YAGO3-10** | .553 | .696 | .601 | .477 | .198 | .324 | .214 | .136 |

Table 10: Link prediction results on WN18 and FB15K obtained from [4,38].

|  | WN18 | | | | FB15K | | | |
|---|---|---|---|---|---|---|---|---|
|  | MRR | Hits@10 | Hits@3 | Hits@1 | MRR | Hits@10 | Hits@3 | Hits@1 |
| DistMult | .822 | .936 | .914 | .728 | .654 | .824 | .733 | .546 |
| ComplEx | .941 | .947 | .936 | .936 | .692 | .840 | .759 | .599 |
| ANALOGY | .942 | .947 | .944 | .939 | .725 | .854 | .785 | .646 |
| R-GCN | .819 | .964 | .929 | .697 | .696 | .842 | .760 | .601 |
| TorusE | .947 | .954 | .950 | .943 | .733 | .832 | .771 | .674 |
| ConvE | .943 | .956 | .946 | .935 | .657 | .831 | .723 | .558 |
| HypER | .951 | .958 | .955 | .947 | .790 | .885 | .829 | .734 |
| SimplE | .942 | .947 | .944 | .939 | .727 | .838 | .773 | .660 |
| TuckER | .953 | .958 | .955 | .949 | .795 | .892 | .833 | .741 |
| QuatE | .950 | .962 | .954 | .944 | .833 | .900 | .859 | .800 |
| CONEX | **.976** | **.980** | **.978** | **.973** | **.872** | **.930** | **.896** | **.837** |

**Link Prediction Results on WN18 and FB15K.** Table 10 reports link prediction results on the WN18 and FB15K benchmark datasets.