# OpenReview forum: "Convolutional Complex Knowledge Graph Embeddings"
_eswc-conferences.org/ESWC/2021/Conference/Research_Track — ESWC 2021 Research_

### Official Review · AnonReviewer2 · 2021-01-11
**An interesting approach to a relevant problem, which is thoroughly evaluated**

**Rating:** 2
**Confidence:** 2
**Impact:** 4
**Design And Technical Quality:** 5

**Review:**

The authors propose a novel link prediction model which combines operations in a complex vector space with 2D convolutions, resulting in a model which exceeds the state-of-the-art performance in many cases, which is robust to overfitting, and which can predict edges that hold a transitive property (a significant problem for the current state-of-the-art). The model is thoroughly evaluated against several baselines and on several benchmark datasets.

The paper is well written, and the theory and results are excellently motivated and explained. The problem that the authors are trying to tackle is also very relevant for the quickly growing field of machine learning on knowledge graphs.

**Anonymity:**

Yes, I would like my review to remain anonymous.

**Reuse And Availability:**

5: Very High

**Strong Points:**

- A good comparison with related work

- A very thorough evaluation against several relevant baselines and on well-known benchmark datasets. Also the consideration to test with varying number of parameters (to evaluate overfitting robustness), and to evaluate on relationship type (symmetric, transitive, etc).

- The proposed approach tackles a problem with the existing state-of-the-art, and does so by extending on these.

- The authors motivate their work well. Same for their explanations of the models behaviour, and their thoughts on the results.

**Subreviewer:**

I submitted this review.

**Weak Points:**

- The authors frequently refer to other papers for information and explanations that preferable are available here as well (in concise form).

- The results are not tested on statistical significance.

---

> ### Author Rebuttal · Authors · 2021-01-30
>
> **We do thank the reviewer for the review and appreciate the time invested.**
>
> **Weak point 1:**
>
> We often referred to other research papers for detailed information. This stems from the space constraint. We will include some definitions of concepts in the appendix on our final version as much as space permits
>
> **Weak point 2:**
>
>  Agreed! We will add statistical tests to the paper.

---

### Official Review · AnonReviewer3 · 2021-01-15
**KGE that combines convolutions with complex-valued embeddings**

**Confidence:** 3
**Impact:** 2
**Design And Technical Quality:** 3

**Review:**

Summary
-------

In this paper, authors propose ConEx (convolutional complex knowledge graph embeddings) - to build the Knowledge Graph Embeddings for relationship detection or link prediction. To be specific, ConEx utilizes the Hadamard product to compose a 2D convolution followed by an affine transformation with a Hermitian inner product. Experiments conducted on the standard link prediction datasets showcase that the proposed ConEx approach is effective.


Comments to Authors
--------------------
1. Comparing to one of the very related approach like Graph Convolutional networks for relation data (Schlichtkrull et al., 2018) will be useful.
2. Complex space C is ill-defined. It will be useful if more details are provided.
3. ConEx has achieved state of the art results on almost four different datasets. However, when comparing the MRR or HIT@10 on most datasets, the results are not statistically significant with other baselines. Does it indicate that trying simpler approaches on complex datasets is still beneficial to reduce complex computations?
4. Although there are different experiments carried out in showcasing the contribution of models on out-of-vocabulary entities and bootstrapping by combing different models. It is surprising that no ablation is performed on the ConEx model to showcase the improvements gained over previous approaches.

Minor Issues
----------

1. Picture depicting the ConEx model architecture is useful and improves readability.

Overall, the paper introduces a yet another method for computing the KGE for link prediction. It combines two previous methods to achieve the state of the art results on different datasets. However, the novelty introduced in the method is minimal and the ablation study is missing to showcase where the exact improvement is achieved.

**Anonymity:**

Yes, I would like my review to remain anonymous.

**Rating:**

-1: Weak Reject

**Reuse And Availability:**

4: High

**Strong Points:**

1. Comprehensive information about Knowledge Graph Embeddings is provided using notation and background setup.
2. Evaluation of ConEx is performed on five standard link prediction datasets.
3. Extensive coverage of previous literature on the Knowledge Graph Embeddings.


**Subreviewer:**

I submitted this review.

**Weak Points:**

1. ConEx contribution is merely seen as scoring function which combines Convolution 2D (Dettmers et al., 2018) with ComplEx (Trouillon et al., 2016) ).
2. Complex space C is ill defined. More details is useful. It is only known that the Hadamard product is used to compose a 2D convolution followed by an affine transformation with a Hermitian
inner product in C.
3. Most of the main details are referred to previous works leaving out many details which are crucial for the paper. For example, details about how convolutions are done between entities etc. Few equations about them or Figure about how computation is made will be useful.

---

> ### Author Rebuttal · Authors · 2021-01-30
>
> **We do thank the reviewer for the review and appreciate the time invested.**
>
> **Point 1: Novelty**
>
> We argue that designing scoring functions is the quintessence of innovative knowledge graph embeddings. For instance,  SimplE (Kazemi and Poole, 2018) presents “a simple enhancement of CP (which we call SimplE) to allow the two embeddings of each entity to be learned dependently”. The paper is still considered a milestone in embeddings (177 citations). Trouillon et al. (2016) is commonly considered a  high-impact paper (799 citations) and builds ComplEx simply by extending DistMult into the complex vector space. Ngyen et al. (2017, 191 citations)’s ConvKB simply removes the reshaping operation in the ConvE model.
>
> **Point 2: ill defined complex space**
>
> We followed the standard definition of the complex vector space  as presented in Trouillon et al. (2016), Trouillon et al. (2017) and Sun et al. (2019). We fail to understand how this commonly accepted definition is ill-defined. We will happily improve our explanation if the reviewer elaborates on what is missing.
>
> **Point 3: Missing details**
>
> We understand this criticism and will provide a figure pertaining to how a convolution is computed between an entity and a relation. Moreover, we will provide as many details as space permits.
>
> Equation 2 elucidates how a 2D convolution is applied on complex-valued embeddings of entities and relations (not between entities). This operation corresponds to applying a 2D convolution on a matrix as shown in (https://bit.ly/3ccebP7).

---

### Official Review · AnonReviewer1 · 2021-01-15
**Impressive results, but a write-up that makes me skeptical**

**Rating:** 1
**Confidence:** 3
**Impact:** 4
**Design And Technical Quality:** 3

**Review:**

The paper proposes a new score function for embedding-based models, which combines ideas from ConvEx and ComplEx. Specifically, the ConvEx decoder is applied to complex embeddings and combined by element-wise multiplication with a ComplEx score.

As the authors note, the convolutional part of the function can be configured to always output 1, so that, in principle, the model subsumes ComplEx.

I have strong objections to many aspects of the write-up, but the fundamental results are impressive. I can attest that it's no easy feat to achieve scores like these in link prediction. If landed on weak reject, but given some discussion, and depending on what the other reviewers say, I may change my opinion.

## Major points

The choice to round performance scores to only two digits makes me skeptical. It is also not clear how this is done. For instance, ComplEx [28] actually achieves 0.475 MRR, which seems like it should be rounded to 0.48. A difference of 0.01 is likely statistically significant, but a difference of 0.001 isn't and both may be the case. The authors should report scores with three or four digits.

The authors make a large number of unsubstantiated claims, for which no or very little experiment evidence is provides. This should be seriously toned down. For instance, on page 6, a claim is made that ConEx is "less prone to vanishing gradients". Presumably less prone than purse ConvEx. This is not evaluation and I seriously doubt that a shallow architecture like ConvEx suffered from vanishing gradients at all. Another example is the claim that ConEx is robust against overfitting (which seems reasonable) and that this is due to the architecture of the score function and it having "sufficient degrees of freedom [...] to minimize the cross-entropy loss." I don't understand how having sufficient degrees of freedom to minimize loss leads to less overfitting. I agree that the model doesn't seem to overfit, even with more parameters, but this may equally be due to the good choice of initialization in pytorch, the choice of the Adam optimizer or the use of dropout.

In general I urge the authors to remove or rewrite any claim for which the experiments do not offer direct and strong proof, and to leave any speculation to the conclusion.

Table 8 is not an ablation study, simply a parameter sweep. An ablation study would be welcome, but it would consist of structurally removing different features from the model (the complex numbers, the multiplication by ComplEx score, training tricks like dropout and label smoothing) to see which feature has the greater impact. This would be a way of substantiating many of the claims in the paper (although they would still have to be phrased more carefully).

I was baffled by the repeated references to bagging (bootstrap aggregating) and the suggestion that this is employed in AlexNet and ResNet. Bagging is not used in these models as far as I'm aware, and what the authors do is also not bagging: it is simply ensembling two models with learnable weights. It could be called stacking, although that usually refers to the case where the models are trained prior to ensembling. Bagging specifically refers to an ensemble of models of the same class, each trained on bootstrapped samples of the training data.

Figure 1 supposedly represents the embedded relations of FB15K-237, but the annotations contain several relations with the suffix "_reverse" which isn't present in this dataset (including the version on the authors' github page).

I like the experiment of table 5. I would prefer that the authors remove the second rows (for the out-of-vocabulary triples, see comment below) and focus more on the properties of the relations for which ConvEx scores particularly well. Are these all transitive relations? If so, what it is about a complex convolution that enables the embeddings to capture such transitivity that a real-valued convolution can't? Does the same hold for FB15k-237? This would really show not just that the method performs well, but also why it performs well.

In general, I suspect that the authors really do have a strong method on their hands, and if so, it would be a shame to publish in the current form. The paper would be made far strong by a thorough analysis of exactly why the method performs so much better and how it achieves this. In particular, a strong ablation study of all features that could account for the performance, and a more detailed investigation by relation could be very illuminating.

## Minor points

- For note marks should follow periods and commas.
- "In the both cases" -> "In both cases", several times.
- I found the repeated use of "ergo" a little jarring, but that may just be me.
- I would not consider KvsAll a data augmentation technique, simply a sampling strategy for negative examples.
- I have read the last paragraph of 5.3 many times, but I simply don't understand what it says.
- Section 5.4: citation [3, 3, 1] contains 3 twice.
- "... while the equivariant representation allows ConEx to effectively integrate interactions captured in the stacked complex-valued embeddings of entities and relations into computations of scores." This sentence makes no sense to me. I don't see how the translation equivariance of convolutions has any relevance to the problem. As far as I understand the reason behind ConvEx the choice of a convolution is just because it is an efficient way to implement a model that is poweful, but not fully connected. The fact that it is wired for equivariance is rather incidental.
-I'd say it's reasonably well known that WN18RR contains entities in the test set that do not occur in the training set, and that these should be left in. It's worth a footnote, certainly, but I don't see the value in repeating the experiment with and without these entities.
- In general, the original FB15k and WN18 should no longer be used. In this case, I see the value in including them , but I would perhaps relegate them to an appendix. It's a little disingenuous top claim superior performance on four out of five datasets when two of those datasets are subsets of two of the others. Most likely all models are modeling the reciprocal relations well, and the authors are simply performing the same experiment twice on the remainder.


**Anonymity:**

No, I would like my review to be deanonymized.

**Reuse And Availability:**

4: High

**Strong Points:**

see above.

**Subreviewer:**

I submitted this review.

**Weak Points:**

see above.

---

> ### Author Rebuttal · Authors · 2021-01-30
>
> **Thank you for the thorough review and the time invested. All minor points will be addressed.**
>
> **Rounding scores**
>
> We will add results with at least 4 digits after the point in the final version of the manuscript. Results without rounding can be obtained using our open-source implementation (https://bit.ly/3pBKICo ).
>
> **Vanishing gradients and overfitting**
>
> Our claim is based on Equation 5, which indicates that the partial derivative of the loss function w.r.t. embeddings can be propagated in two ways, namely, via conv(.,.) and via ComplEx(.,.,.). Hence, overall, the partial derivative are at most as likely to be zero as  in conv() or ComplEx(). Still, we do get the reviewer’s perspective and will tone down that claim. Pertaining to overfitting,  Dettmers et al. (2017) and Balazevic et al. (2018) have shown the usefulness of convolutions to prevent overfitting. Still, we will tone down this claim and add an ablation study to quantify the robustness of ConEx against overfitting by removing dropout and batch normalization.
>
> **Ablation study**
>
> We agree with the critique of the reviewer and plan to conduct a more extensive ablation analysis. We thank the reviewer for the suggestions pertaining to the design thereof. The results will be available in the final version of the manuscript and on the project page.
>
> **Bootstrapping**
>
> We do agree that what we do can be regarded as ensembling indeed. According to Pattern Recognition and Machine Learning (Bishop, 2006), the definition of bootstrapping due to (Breiman, 1996) does not require the bootstrap datasets to be different. Hence, formally, we are indeed carrying out bootstrapping. Still, we’d be fine with calling it ensembling for the sake of simplicity. We do not have any learnable parameter at averaging predictions  (https://bit.ly/3iZ8Gox)
>
>
> **Link prediction per relation on Table 5**
>
> We will reshape the table 5 to provide more information pertaining to types of relations as the reviewer suggested. We performed the link prediction per relation evaluation on other datasets as well. Due to the space constraint, we did not include results. Link prediction per relation on FB15K-237 can be easily obtained (https://bit.ly/2Yh7Ycu).
>
> **The equivariant representation and convolutions**
>
> The parameter sharing ability endows convolutions with the equivariant representation property (see chapter 9.2 in Deep Learning Goodfellow et al.). The usefulness of this property in detecting interactions on images is quite well known. In our work, we aimed to benefit from this property to detect interactions between complex-valued embeddings of entities and relations. We hope this elucidates our motivation behind using convolutions on complex-valued embeddings.
>
> **Using FB15k and WN18**
>
> We’ll happily move these results to the appendix.

---

> > ### Comment · AnonReviewer1 · 2021-02-05
> > **Review revised**
> >
> > Given the authors' replies, I have changed my review to a weak accept. I would urge the authors to include results that were not peer-reviewed only in an appendix, and clearly marked as such.
> >
> > Some points (such as the suffixes in Figure 1) were not addressed. I trust that the authors will also address all points not explicitly mentioned in their final version.
> >
> > If the authors can find the time, I recommend submitting an implementation of their score function to the libKGE project by pull request, so that the performance of the model may be evaluated in a more extensive experiment (which is outside the scope of the current research).

---

### Official Review · AnonReviewer4 · 2021-01-15
**Novel extension to ComplEx with excellent presentation and evaluation**

**Rating:** 2
**Confidence:** 3
**Impact:** 4
**Design And Technical Quality:** 4

**Review:**

The work presents an extension to the ComplEx graph embedding method that improves the shortcomings of ComplEx in terms of transitive relations.

The paper does an excellent job at summarizing related work and highlighting the differences with the proposed method. The motivation and design choices are clearly presented and easy to follow. The presented methodology seems novel and impactful. The evaluation is extensive (five datasets) and convincing (comparison to SOTA models). Code and models are available online.

The shortcomings of the paper are minor and mostly cosmetic:
* Figure 1 needs more elaboration: What are the conclusions drawn from it?
* Missing citation and typo in 5.3 "Ruffinelli et al."
* In Tables 4 and 7 it is not explained what an underlined number represents.
* Some of the references seem to cite the arXiv versions, rather than the conference versions (e.g., 3 and 13).

**Anonymity:**

Yes, I would like my review to remain anonymous.

**Reuse And Availability:**

5: Very High

**Subreviewer:**

I submitted this review.

---

> ### Author Rebuttal · Authors · 2021-01-30
>
> **We do thank the reviewer for the review and appreciate the time invested.**
>
> We will revise the references and fix the typos. Moreover, we agree with the reviewer that figure 1 needs more elaboration. This will be done in our final version.

---

> > ### Comment · AnonReviewer4 · 2021-02-08
> > **Acknowledged**
> >
> > Thanks!

---

### Official Review · AnonReviewer5 · 2021-01-17
**The authors proposed ConEx as the new algorithm contribution to the family of knowlege embedding. The proposed method can outperform on some baseline datasets.**

**Rating:** 2
**Confidence:** 2
**Impact:** 4
**Design And Technical Quality:** 3

**Review:**

The authors provided a novel algorithmic approach called ConEx to deal the issues of continuous vector representations of knowledge graphs. ConEx outperforms baseline methods on WN18 and FB15K, but not on WN18RR and FB15K-237. But the bootstrapping combinations are promising (Table 6).


**Anonymity:**

Yes, I would like my review to remain anonymous.

**Reuse And Availability:**

3: Medium

**Strong Points:**

- novel algorithmic contribution
- systematic evaluation with baseline methods and benchmark datasets
- parameter tuning
- ConEx outperforms on many important relationships (table 5)

**Subreviewer:**

I submitted this review.

**Weak Points:**

- performance improvement is around 1%

---

> ### Author Rebuttal · Authors · 2021-01-30
>
> We do thank the reviewer for the review and appreciate the time invested.

---

### Decision · Program_Chairs · 2021-02-23

**Decision:**

Accept with shepherding

**Comment:**

For our meta-review, we have paid more attention to AnonReviewer1, AnonReviewer3 and  AnonReviewer2 because of the greater depth of their reviews.

Overall, the reviewers agree that this is an interesting continuation of the well-known ComplEx embedding method.
The results are generally good, but several concerns remain.  Because of this we will accept this paper with shepherding, meaning that if the requested improvements are made, the paper can be accepted.

The aspects which would have to be improved are the following:
1. Statistical significance of your results.

Currently the results are presented without sufficient measures of statistical significance. This request is triggered by two views. First, you claim differences which might be due to rounding because only limited significant figures are included. But then, even if more significant figures would be included it is not clear whether that difference would be significant, and not just be caused by random chance.

2. The ablations study as requested by AnonReviewer1 to which you already agreed.

We agree that an ablation study would be one way to substantiate several of the claims the reviewers are raising regarding the work.

---

> ### Comment · Area_Chair2 · 2021-03-16
> **Paper accepted**
>
> An update of the paper was provided and the points raised in the shepherding are addressed sufficiently.
>
> A remaining concern is how 2 initializations can allow the computation a confidence interval in the ablation test. Surely you need n >= 3 to even compute the variance? Please explain with a bit more detail how the CI is computed (just one sentence or a footnote).